# A Free Lunch from the Noise:
# Provable and Practical Exploration for Representation Learning

**Tongzheng Ren** [1, 2, *]          **Tianjun Zhang** [3, *]          **Csaba Szepesvári** [4, 5]          **Bo Dai** [2]

[1]Department of Computer Science, UT Austin
[2]Google Research, Brain Team
[3] Department of EECS, UC Berkeley
[4]DeepMind
[5]Department of Computer Science, University of Alberta

## Abstract

Representation learning lies at the heart of the empirical success of deep learning for dealing with the curse of dimensionality. However, the power of representation learning has not been fully exploited yet in reinforcement learning (RL), due to **i)**, the trade-off between expressiveness and tractability; and **ii)**, the coupling between exploration and representation learning. In this paper, we first reveal the fact that under some noise assumption in the stochastic control model, we can obtain the linear spectral feature of its corresponding Markov transition operator in closed-form *for free*. Based on this observation, we propose *Spectral Dynamics Embedding (SPEDE)*, which breaks the trade-off and completes optimistic exploration for representation learning by exploiting the structure of the noise. We provide rigorous theoretical analysis of SPEDE, and demonstrate the practical superior performance over the existing state-of-the-art empirical algorithms on several benchmarks.

## 1 INTRODUCTION

Reinforcement learning (RL) dedicates to solve the sequential decision making problem, where an agent is interacting with an *unknown* environment to find the best policy that maximizes the expected cumulative rewards [Sutton and Barto, 2018]. It is known that the tabular algorithms direct controlling over the original state and action in Markov decision processes (MDPs) achieve the minimax-optimal regret depending on the cardinality of the state and action space [Jaksch et al., 2010, Azar et al., 2017, Jin et al., 2018]. However, these algorithms become computationally intractable for the real-world problems with an enormous number of states. Learning with function approximations

upon *good* representation is a natural idea to tackle such computational issue, which has already demonstrated its effectiveness in the success of deep learning [Bengio et al., 2013]. In fact, representation learning lies at the heart of the empirical successes of deep RL in video games [Mnih et al., 2013], robotics [Levine et al., 2016], Go [Silver et al., 2017] to name a few. Meanwhile, the importance and benefits of the representation in RL is rigorously justified [Jin et al., 2020, Yang and Wang, 2020], which quantifies the regret in terms of the dimension of the *known* representation based on a subclass in MDPs [Puterman, 2014]. A natural question raises:

*How to design **provably efficient** and **practical** algorithm for representation learning in RL?*

Here, by "provably efficient" we mean the sample complexity of the algorithm can be rigorously characterized only in terms of the complexity of representation class, without explicit dependency on the number of states and actions, while by "practical" we mean the algorithm can be implemented and deployed for the real-world applications. Therefore, we not only require the representation learned is expressive enough for handling complex practical environments, but also require the operations in the algorithm tractable and computation/memory efficient. The major difficulty of this question lies in two-fold:

- **i)** The *trade-off* between the expressiveness and the tractability in the design of the representations;
- **ii)** The learning of representation is intimately *coupled* with exploration.

Specifically, a desired representation should be sufficiently expressive to capture the practical dynamic systems, while still computationally tractable. However, in general, expressive representation leads to complicated optimization in

---

* Equal Contribution

For a formal definition of expressiveness, see [Agarwal et al., 2020a].

*Accepted for the 38th Conference on Uncertainty in Artificial Intelligence* (UAI 2022).

learning. For example, the representation in the linear MDP is *exponential stronger* than the latent variable MDPs in terms of expressiveness [Agarwal et al., 2020a]. However, its representation learning depends on either a MLE oracle that is computationally intractable due to the constraint on the regularity of conditional density [Agarwal et al., 2020a], or an optimization oracle that can solve complicated constrained $\mathrm{min}$-$\mathrm{max}$-$\mathrm{min}$-$\mathrm{max}$ optimization [Modi et al., 2021]. On the other hand, Misra et al. [2020] considers the representation introduced by an encoder in block MDP [Du et al., 2019], in which the learning problem can be completed by a regression, but with the payoff that the representations in block MDP is even weaker than the latent variable MDP [Agarwal et al., 2020a].

Meanwhile, the coupling of the representation learning and exploration also induces the difficulty in *practical* algorithm design and analysis. Specifically, one cannot learn a precise representation without enough experiences from a comprehensive exploration, while the exploration depends on a reliable estimation of the representation. Most of the known results depends on a policy-cover-based exploration [Du et al., 2019, Misra et al., 2020, Agarwal et al., 2020a, Modi et al., 2021], which maintains and samples a set of policies during training for systematic exploration, that significantly increases the computation and memory cost in implementation.

In this work, we propose *Spectral Dynamics Embedding (SPEDE)*, dealing with the aforementioned difficulties appropriately, and thus, answering the question affirmatively. SPEDE is established on a connection between the stochastic control dynamics [Osband and Van Roy, 2014, Kakade et al., 2020] and linear MDPs in Section 3. Specifically, by exploiting the property of the *noise* in the stochastic control dynamics, we can recover the factorization of its corresponding Markov transition operator in closed-form *without extra computation*. This equivalency immediately overcomes the computational intractability in the linear MDP estimation via the corresponding control dynamics form, and thus, breaks the trade-off between expressiveness and tractability. Meanwhile, as a byproduct, the linear MDP reformulation also introduce efficient planning for optimal policy in nonlinear control through the linear sufficient feature from the spectral space of Markov operator, while in most model-based RL, planning is conducted by treating learned model as simulator, and thus, is inefficient and suboptimal.

More importantly, the two faces of one model also provide the opportunity to tackle the coupling between representation learning and exploration. The optimism in the face of uncertainty principle can be easily implemented through Thompson sampling w.r.t. the stochastic nonlinear dynamics, which leads to the posterior of representations implicitly, while bypasses the unidentifiability issue in directly characterizing the representation, therefore, can be theoretically justified.

We rigorously characterize the statistical property of SPEDE in terms of regret w.r.t. the complexity of representation class in Section 4, without explicit dependence on the size of raw state space and action space. With the established unified view, our results generalize online control [Kakade et al., 2020] and linear MDP [Jin et al., 2020] beyond *known* features. We finally demonstrate the superiority of SPEDE on the MuJoCo benchmarks in Section 5. It significantly outperforms the empirical state-of-the-art RL algorithms. To our knowledge, SPEDE is the first representation learning algorithm achieving statistical, computational, and memory efficiency with sufficient expressiveness.

## 1.1 RELATED WORK

There have been many great attempts on **algorithmic representation learning** in RL for different purposes, *e.g.*, bisimulation [Ferns et al., 2004, Gelada et al., 2019], reconstruction [Hafner et al., 2019]. Recently, there are also several works considering the spectral features based on decomposing different variants of the transition operator, including successor features [Dayan, 1993, Kulkarni et al., 2016], proto-value functions [Mahadevan and Maggioni, 2007, Wu et al., 2018], spectral state-aggregation [Duan et al., 2018, Zhang and Wang, 2019], and contrastive fourier features [Nachum and Yang, 2021]. These works are highly-related to the proposed SPEDE. Besides these features focus on *state-only* representation, the major differences between SPEDE and these spectral features lie in **i)**, the target operators in existing spectral features are *state-state* transition, which cancel the effect of action; **ii)**, the target operators are estimated based on empirical data from a *fixed behavior policy* under the implicit assumption that the estimated operator is *uniformly accurate*, ignoring the major difficulty in exploration, while SPEDE carefully designed the systematic exploration with theoretical guarantee; **iii)**, most of the existing spectral features rely on *explicitly* decomposition of the operators, while SPEDE obtains the spectral *for free*.

Turning to the **theoretically-justified representation learning with online exploration**, a large body of effort focuses on the policy-cover-based exploration [Du et al., 2019, Misra et al., 2020, Agarwal et al., 2020a, Modi et al., 2021]. The major difficulty impedes their practical application is the computation and memory cost: the policy-cover-based exploration requires a set of exploratory polices to be maintained and sampled from during training, which can be extremely expensive. Uehara et al. [2021] introduced a UCB mechanism that can enforce exploration without the requirements on maintaining the policy cover. However, the algorithm requires an MLE oracle for unnormalized conditional statistical model, which still prevents us from applying the algorithm in practice until recent attempt [Zhang

et al., 2022] using contrastive learning to replace the MLE.

Another two related lines of research are **model-based RL** and **online control**, which are commonly known overlapped but separate communities considering different formulations of the dynamics. Our finding bridges these two communities by establishing the equivalency between standard models that are widely considered in the corresponding communities. Osband and Van Roy [2014] and Kakade et al. [2020] are the most related to our work in each community. These models generalize their corresponded linear models, *i.e.*, Jin et al. [2020] and Cohen et al. [2019], with general nonlinear model and kernel function within a known RKHS, respectively. The regret of the optimistic (pessimistic) algorithm has been carefully characterized for these models. However, both of the proposed algorithms in Osband and Van Roy [2014] and Kakade et al. [2020] require a planning oracle to seek the optimal policy, which might be computationally intractable. In SPEDE, this is easily handled in the equivalent linear MDP.

## 2 PRELIMINARIES

Markov Decision Process (MDP) is one of the most standard models studied in the reinforcement learning that can be denoted by the tuple $\mathcal{M} = (\mathcal{S}, \mathcal{A}, r, T, \rho, H)$, where $\mathcal{S}$ is the state space, $\mathcal{A}$ is the action space, $r : \mathcal{S} \times \mathcal{A} \to \mathbb{R}^+$ is the reward function (where $\mathbb{R}^+$ denotes the set of non-negative real numbers), $T : \mathcal{S} \times \mathcal{A} \to \Delta(\mathcal{S})$ is the transition and $\rho$ is an initial state distribution and $H$ is the horizon (*i.e.*, the length of each episode). A (potentially non-stationary) policy $\pi$ can be defined as $\{\pi_h\}_{h \in [H]}$ where $\pi_h : \mathcal{S} \to \Delta(\mathcal{A}), \forall h \in [H]$. Following the standard notation, we define the value function $V_h^\pi(s_h) := \mathbb{E}_{T,\pi}\left[\sum_{t=h}^{H-1} r(s_t, a_t)|s_h = s\right]$ and the action-value function (*i.e.*, the $Q$ function) $Q_h^\pi(s_h, a_h) = \mathbb{E}_{T,\pi}\left[\sum_{t=h}^{H-1} r(s_t, a_t)|s_h = s, a_h = a\right]$, which are the expected cumulative rewards under transition $T$ when executing policy $\pi$ starting from $s_h$ and $(s_h, a_h)$. With these two definitions at hand, it is straightforward to show the following Bellman equation:

$$Q_h^\pi(s_h, a_h) = r(s_h, a_h) + \mathbb{E}_{s_{h+1} \sim T(\cdot|s_h, a_h)}\left[V_{h+1}^\pi(s_{h+1})\right].$$

Most of RL algorithms aim at finding the optimal policy $\pi^* = \arg\max_\pi \mathbb{E}_{s \sim \rho}\left[V_0^\pi(s)\right]$ under MDPs. It is well known that in the tabular setting when the state space and action space are finite, we can provably identify the optimal policy with both sample-efficient and computational-efficient optimism-based methods [*e.g.* Azar et al., 2017]

---

In general, the reward can be stochastic. Here for simplicity we assume the reward is deterministic and known throughout the paper, which is a common assumption in the literature [*e.g.*, Jin et al., 2018, 2020, Kakade et al., 2020].

Our method can be generalized to infinite horizon case, see Section 3.2 for the detail.

with the complexity proportion to $\text{poly}(|\mathcal{S}|, |\mathcal{A}|)$. However, in practice, the cardinality of state and action space can be large or even infinite. Hence, we need to incorporate function approximation into the learning algorithm when we deal with such cases. The linear MDP [Jin et al., 2020] or low-rank MDP [Agarwal et al., 2020a, Modi et al., 2021] is the most well-known MDP class that can incorporate linear function approximation with theoretical guarantee, thanks to the following assumption on the transition and reward:

$$T(s'|s, a) = \langle \phi(s, a), \mu(s') \rangle_{\mathcal{H}}, \quad r(s, a) = \langle \phi(s, a), \theta \rangle_{\mathcal{H}}, \tag{1}$$

where $\phi : \mathcal{S} \times \mathcal{A} \to \mathcal{H}$, $\mu : \mathcal{S} \to \mathcal{H}$ are two feature maps and $\mathcal{H}$ is a Hilbert space. The most essential observation for them is that, $Q_h^\pi(s, a)$ for any policy $\pi$ is linear w.r.t $\phi(s_h, a_h)$, due to the following observation [Jin et al., 2020]:

$$Q_h^\pi(s, a) = r(s, a) + \int V_{h+1}^\pi(s_{h+1}) T(s_{h+1}|s_h, a_h)\, \mathrm{d}s_{h+1}$$

$$= \left\langle \phi(s_h, a_h), \theta + \int V_{h+1}^\pi(s_{h+1})\mu(s_{h+1})\, \mathrm{d}s_{h+1} \right\rangle_{\mathcal{H}}. \tag{2}$$

Therefore, $\phi$ serves as a sufficient representation for the estimation of $Q_h^\pi$, that can provide uncertainty estimation with standard linear model analysis and eventually lead to sample-efficient learning when $\phi$ is fixed and known to the agent [see Theorem 3.1 in Jin et al., 2020]. However, we in general do not have such representations in advance and we need to learn the representation from the data, which constraints the applicability of the algorithms derived with fixed and known representation.

**Remark (Model-based RL vs. RL with Representation):** We would like to emphasize that, although most of the existing representation learning methods need to learn the transition [Du et al., 2019, Misra et al., 2020, Agarwal et al., 2020a, Uehara et al., 2021], RL with representation learning is related but perpendicular to the concept of model-based RL. The major difference lies in how to use the learned transition for planning (*i.e.* finding the optimal policy). In vanilla model-based RL methods [*e.g.*, Sutton, 1990, Chua et al., 2018, Kurutach et al., 2018], the learned transition is played as a simulator generating samples for policy improvement; while in representation-based RL, the representation is extracted from the learned transition to compose the policy explicitly, which is significantly efficient comparing to the model-based RL methods.

---

One exception is the tabular MDP, where we can choose $\phi : \mathcal{S} \times \mathcal{A} \to \mathbb{R}^{|\mathcal{S}||\mathcal{A}|}$ that each state-action pair has exclusive one non-zero element and $\mu : \mathcal{S} \to \mathbb{R}^{|\mathcal{S}||\mathcal{A}|}$ correspondingly defined to make (1) hold.

# 3 SPECTRAL DYNAMICS EMBEDDING

It is naturally to consider how to perform sample-efficient representation learning (and hence sample-efficient reinforcement learning) that satisfies (1) in an online manner. The most straightforward idea is performing the maximum likelihood estimation (MLE) in the representation space [*e.g.*, Agarwal et al., 2020a]. Unfortunately, for general cases, such MLE is intractable, due to the constraints on the regularity of marginal distribution (*i.e.*, $\langle \phi(s,a), \int_{s'} \mu(s') \, \mathrm{d}s' \rangle = 1$) for all $(s,a) \in \mathcal{S} \times \mathcal{A}$. Moreover, even we can perform MLE for certain cases (for example, the block MDP), as the representation is estimated from the data, which can be inaccurate, most of the existing work apply the policy cover technique [Du et al., 2019, Misra et al., 2020, Agarwal et al., 2020a, Modi et al., 2021] to enforce exploration. However, such procedures can be both computational and memory expensive when we need amounts of exploratory policy to guarantee the coverage of whole state space, which makes it not a practical choice.

To overcome these issues, we introduce Spectral Dynamics Embedding (SPEDE), which leverages the noise structure to provide a simple but provable efficient and practical algorithm for representation learning in RL. We first introduce our key observation, which induces the equivalency between linear MDP and stochastic nonlinear control.

## 3.1 KEY OBSERVATION

Our fundamental observation is that, the density of isotropic Gaussian distribution can be expressed as the inner product of two feature maps, thanks to the reproducing property and the random Fourier transform of the Gaussian kernel [Rahimi and Recht, 2007]:

$$
\begin{aligned}
\phi(x|\mu, \sigma^2 I) &\propto \exp\left(-\frac{\|x-\mu\|^2}{2\sigma^2}\right) \\
&= \langle k(x, \cdot), k(\mu, \cdot) \rangle_{\mathcal{H}} \quad \textit{(Reproducing Property)} \\
&\qquad\qquad\qquad\qquad\qquad\qquad (3) \\
&= \langle \varphi(x, \omega, b), \varphi(\mu, \omega, b) \rangle_{p(\omega, b)} \quad \textit{(Random Fourier)}, \\
&\qquad\qquad\qquad\qquad\qquad\qquad (4)
\end{aligned}
$$

where $k(\cdot, \cdot)$ is the Gaussian kernel with bandwidth $\sigma$: $k(x,y) = \exp\left(-\frac{\|x-y\|_2^2}{2\sigma^2}\right)$, $\mathcal{H}$ is the Reproducing Kernel Hilbert Space (RKHS) associated with $k$, $\varphi(x, \omega, b) = \sqrt{2}\cos(\omega^\top x + b)$, $\langle f, g \rangle_p = \mathbb{E}_{p(x)}[f(x)g(x)]$ and $p(\omega, b) = \mathcal{N}(\omega; 0, 1/\sigma^2 I) \cdot \mathcal{U}(b; [0, 2\pi])$ with $\mathcal{N}$ and $\mathcal{U}$ denoting Gaussian and Uniform distribution, respectively.

---

We provide a brief review on the related definitions in Appendix A.

Consider the general transition dynamics,

$$s' = f^*(s,a) + \epsilon, \quad \epsilon \sim \mathcal{N}(0, \sigma^2), \qquad (5)$$

or equivalently $\quad T(s'|s,a) \propto \exp\left(-\frac{\|s' - f^*(s,a)\|^2}{2\sigma^2}\right),$

$$(6)$$

which is a widely used setup in the empirical model-based reinforcement learning [*e.g.*, Chua et al., 2018, Kurutach et al., 2018, Clavera et al., 2018, Wang et al., 2019], and the online (non)-linear control [*e.g.*, Abbasi-Yadkori and Szepesvári, 2011, Mania et al., 2019, 2020, Simchowitz and Foster, 2020, Kakade et al., 2020]. Here $s \in \mathbb{R}^d$, $a \in \mathcal{A}$ that can be continuous and $f^*$ is a dynamic function.

By applying the reproducing property (3) or random Fourier transform (4) for the transition dynamics (5), we can obtain the feature $\phi$ and $\mu$ satisfies (1) *for free*. Specifically, taking the reproducing property as an example, we have that

$$T(s'|s,a) = \langle k(f^*(s,a), \cdot), (2\pi\sigma^2)^{-d/2} k(s', \cdot)\rangle_{\mathcal{H}}, \quad (7)$$

which means the problem (5) is indeed a linear MDP with $\phi(s,a) = k(f^*(s,a), \cdot)$ and $\mu(s') = (2\pi\sigma^2)^{-d/2}k(s', \cdot)$. Following (2), we know $Q(s,a)$ is in the linear span of the $\phi(s,a)$ that is transformed from $f^*(s,a)$. Therefore, finding a good representation of $Q(s,a)$ is equivalent to finding a good estimation of $f^*$. In the next section, we will show that, with the well-known optimism in the face of uncertainty (OFU) principle, we can estimate $f^*$ in an online manner with a both sample-efficient in terms of regret and computational-efficient algorithm.

**Remark (Computation-free Factorizable Noise Model):** We remark that, similar observations also hold for large amounts of distributions, *e.g.*, the Laplace and Cauchy distribution. We refer the interested reader to Table 1 in Dai et al. [2014] for the known transformation of kernels and features. Here we focus on the Gaussian noise.

**Remark (Reward Factorization):** In the definition of linear MDP (1), the reward function $r(s,a)$ should also have the ability to be linearly represented by $\phi(s,a)$. This can be implemented by augmenting $[\phi(s,a), r(s,a)]$ as the new representation, therefore, we neglect the reward function throughout the paper.

## 3.2 PRACTICAL ALGORITHM DESCRIPTION

Here, we introduce a generic Thompson Sampling (TS) type algorithm in Algorithm 1 based on the OFU principle that leverage our observation at the previous section. At the beginning, we provide a prior distribution $\mathbb{P}(f)$ that reflects our prior knowledge on $f^*$. Then for each episode, we draw a $f$ from the posterior, find the optimal policy with $f$ using the planning algorithm, execute this policy and eventually

---

**Algorithm 1** Thompson Sampling (TS) Algorithm

---

**Require:** Number of Episodes $K$, Prior Distribution $\mathbb{P}(f)$, Reward Function $r(s, a)$.

1: Initialize the history set $\mathcal{H}_0 = \emptyset$.
2: **for** episodes $k = 1, 2, \cdots$ **do**
3:     Sample $f_k \sim \mathbb{P}(f | \mathcal{H}_k)$. ▷ Draw the Representation.
4:     Find the optimal policy $\pi_k$ on $f_k$ with Algorithm 2. ▷ Planning with $f_k$.
5:     **for** steps $h = 0, 1, \cdots, H - 1$ **do** ▷ Executing $\pi_k$.
6:         Execute $a_h^k \sim \pi_k^h(s_h^k)$.
7:         Observe $s_{h+1}$.
8:     **end for**
9:     Set $\mathcal{H}_k = \mathcal{H}_{k-1} \cup \{(s_h^k, a_h^k, s_{h+1}^k)\}_{h=0}^{H-1}$. ▷ Update the History.
10: **end for**

---

**Algorithm 2** Planning with Dynamic Programming

---

**Require:** Transition Model $f$, Reward Function $r(s, a)$.

1: Initialize $\phi(s, a)$, $\mu(s')$ with (3) or (4). $V_H(s) = 0, \forall s$.
2: **for** steps $h = H - 1, H - 2, \cdots, 0$ **do**
3:     Compute

$$Q_h(s, a) = r(s, a) + \langle \phi(s, a), \int V_{h+1}(s')\mu(s')\,\mathrm{d}s' \rangle_{\mathcal{H}}.$$

                                  ▷ Bellman Update.
4:     Set $V_h(s) = \max_a Q_h(s, a)$, $\pi_h(s) = \arg\max_a Q_h(s, a)$. ▷ Choose the Optimal Policy.
5: **end for**
6: **return** $\{\pi_h\}_{h=0}^{H-1}$.

---

inference the posterior with the new observation. Notice that, we choose the policy optimistically with an *sampled* $f$, which enforces the exploration following the principle of OFU. Meanwhile, we only learn the dynamic with posterior inference and directly obtain the representation with (3) or (4), which avoids additional error from the representation learning step. As all of our data is collected with $f^*$, our posterior will shrink to a point mass of $f^*$, which guarantees we can identify good representation and good policy with sufficient number of data.

One significant part of SPEDE is the computational-efficient planning with $f_k$, thanks to the linear MDP formulation (7). Prior work assumes an oracle [*e.g.*, Kakade et al., 2020] for such planning problem, but little is known on how to provably perform such planning efficiently. Notice that, with the feature $\phi(s, a)$ defined via (3) and (4), we know that $Q_h^\pi(s, a)$ is exactly linear in $\phi(s, a), \forall h, \pi$. Hence, we can perform a dynamic programming style algorithm that calculates $Q_h^\pi(s, a)$ with the given feature $\phi(s, a)$, and then greedily select the action at each level $h$, which is simple yet efficient. It is straightforward to show that the policy obtained with this dynamic programming algorithm is optimal by induction. We illustrate the detailed algorithm in Algorithm 2.

### 3.2.1 Implementation Details

In such a planning algorithm, we need to maintain the posterior of $f$ and calculate the term $\int V_{h+1}(s')\mu(s')\,\mathrm{d}s'$ and take the maximum of $Q_h(s, a)$ over $a$, which can be problematic. We will provide more discussion on this issue below.

**Posterior Sampling** The exact posterior inference can be hard if $f^*$ does not lie in simple function class (*e.g.*, linear function class) or has some derived property (*e.g.*, conjugacy), so in practice we apply the existing mature

approximate inference methods like Markov Chain Monte Carlo (MCMC) [*e.g.*, Neal et al., 2011] and variational inference [see, Blei et al., 2017]. In our implementation, we used stochastic gradient langevin dynamics [Welling and Teh, 2011, Cheng and Bartlett, 2018] to train an ensemble of models for posterior approximation.

**Large State and Action Space** In general, we need to handle the case when the number of states and actions can be large, or even infinite. Notice that, when the state space is large, we can estimate the term $\int V_{h+1}(s')\mu(s')\,\mathrm{d}s'$ with regression based method using the samples from $f$ [Antos et al., 2008]. For the continuous action space, we can apply principled policy optimization methods [*e.g.*, Agarwal et al., 2020b] with an energy-based model (EBM) parametrized policy [Nachum et al., 2017, Dai et al., 2018], treat the linear $Q^\pi(s, a)$ as the gradient and perform mirror descent and eventually obtain the optimal policy. However, this is at the cost of an additional sampling step from the EBM policy. In practice, we introduce a Gaussian policy and perform soft actor-critic [Haarnoja et al., 2018] policy update, which already provides good empirical performance. To sum up, for large state and action cases, we learn the critic in the learned representation space by regression, and obtain the Gaussian parametrized actor with SAC policy update step, in Line 3 and 4 in Algorithm 2, respectively.

**Infinite Horizon Case** Our algorithm can be provably extended to the infinite horizon case with specific termination condition for each episode [*e.g.*, see Jaksch et al., 2010]. In practice, for the planning part we can solve the linear fixed-point equation with the feature $\phi(s, a)$ using the popular algorithms like Fitted $Q$-iteration (FQI) [Antos et al., 2008] or dual embedding [Dai et al., 2018]. that still guarantees to find the optimal policy.

# 4 THEORETICAL GUARANTEES

In this section, we provide theoretical justification for SPEDE, showing that SPEDE can identify informative representation and as a result, near-optimal policy in a sample-efficient way.

We first define the notation of regret. Assume at episode $k$, the learner chooses the policy $\pi_k$ and observes a sequence $\{(s_h^k, a_h^k)\}_{h=0}^{H-1}$. We define the regret of the first $K$ episodes (and define $T := KH$) as:

$$\text{Regret}(K) := \sum_{k \in [K]} \left[ V_0^*(s_0^k) - V_0^{\pi_k}(s_0^k) \right] \quad (8)$$

The regret measures the sample complexity of the representation learning in RL. We want to provide a regret upper bound that is sublinear in $T$. When $T$ increases, we collect more data that can help us build a much more accurate estimation on the representation, which should decrease the per-step regret and make the overall regret scale sublinear in $T$. As we consider the Thompson Sampling algorithm, we would like to study the expected regret $\mathbb{E}_{\mathbb{P}(f)}[\text{Regret}(K)]$, which takes the prior $\mathbb{P}(f)$ into account.

## 4.1 ASSUMPTIONS

Before we start, we first state the assumptions we use to derive our theoretical results.

We assume the reward is bounded, which is common in the literature [*e.g.* Azar et al., 2017, Jin et al., 2018, 2020].

**Assumption 1** (Bounded Reward). $r(s, a) \in [0, 1]$, $\forall (s, a) \in \mathcal{S} \times \mathcal{A}$.

In practice, we generally approximate $f^*$ with some complicated function approximators, so we focus on the setting where we want to find $f^*$ from a general function class $\mathcal{F}$ This is important for MuJoCo dynamics modeling, which have complicated transitions over angle, angular velocity and torque of the agent in the raw state. We first state some necessary definitions and assumptions on $\mathcal{F}$.

**Definition 1** ($\ell_2$-norm of functions). *Define* $\|f\|_2 := \max_{(s,a) \in \mathcal{S} \times \mathcal{A}} \|f(s, a)\|_2$. *Notice that it is not the commonly used $\ell_2$ norm for the function, but it suits our purpose well.*

**Assumption 2** (Bounded Output). *We assume that* $\|f\|_2 \leq C, \forall f \in \mathcal{F}$.

**Assumption 3** (Realizability). *We assume the ground truth dynamic function* $f^* \in \mathcal{F}$.

We then define the notion of covering number, which will be helpful in our algorithm derivation.

**Definition 2** (Covering Number [Wainwright, 2019]). *An $\epsilon$-cover of $\mathcal{F}$ with respect to a metric $\rho$ is a set $\{f_i\}_{i \in [n]} \subseteq \mathcal{F}$, such that $\forall f \in \mathcal{F}$, there exists $i \in [n]$, $\rho(f, f_i) \leq \epsilon$. The $\epsilon$-covering number is the cardinality of the smallest $\epsilon$-cover, denoted as $\mathcal{N}(\mathcal{F}, \epsilon, \rho)$.*

**Assumption 4** (Bounded Covering Number). *We assume that* $\mathcal{N}(\mathcal{F}, \epsilon, \|\cdot\|_2) < \infty, \forall \epsilon > 0$.

**Remark** Basically, Assumption 2 means the the transition dynamic never pushes the state far from the origin, which holds widely in practice. Assumption 3 guarantees that we can find the exact $f^*$ in $\mathcal{F}$, or we will always suffer from the error induced by model mismatch. Assumption 4 ensures that we can estimate $f^*$ with small error when we have sufficient number of observations.

Besides the bounded covering number, we also need an additional assumption on bounded eluder dimension, which is defined in the following:

**Definition 3** ($\epsilon$-dependency [Osband and Van Roy, 2014]). *A state-action pair $(s, a) \in \mathcal{S} \times \mathcal{A}$ is $\epsilon$-dependent on $\{(s_i, a_i)\}_{i \in [n]} \subseteq \mathcal{S} \times \mathcal{A}$ with respect to $\mathcal{F}$, if $\forall f, \tilde{f} \in \mathcal{F}$ satisfying $\sqrt{\sum_{i \in [n]} \|f(s_i, a_i) - \tilde{f}(s_i, a_i)\|_2^2} \leq \epsilon$ satisfies that $\|f(s, a) - \tilde{f}(s, a)\|_2 \leq \epsilon$. Furthermore, $(s, a)$ is said to be $\epsilon$-independent of $\{(s_i, a_i)\}_{i \in [n]}$ with respect to $\mathcal{F}$ if it is not $\epsilon$-dependent on $\{(s_i, a_i)\}_{i \in [n]}$.*

**Definition 4** (Eluder Dimension [Osband and Van Roy, 2014]). *We define the eluder dimension $\dim_E(\mathcal{F}, \epsilon)$ as the length $d$ of the longest sequence of elements in $\mathcal{S} \times \mathcal{A}$, such that $\exists \epsilon' \geq \epsilon$, every element is $\epsilon'$-independent of its predecessors.*

**Remark** Intuitively, eluder dimension illustrates the number of samples we need to make our prediction on unseen data accurate. If the eluder dimension is unbounded, then we cannot make any meaningful prediction on unseen data even with large amounts of collected samples. Hence, to make the learning possible, we need the following bounded eluder dimension assumption.

**Assumption 5** (Bounded Eluder Dimension). *We assume* $\dim_E(\mathcal{F}, \epsilon) < \infty, \forall \epsilon > 0$.

## 4.2 MAIN RESULT

**Theorem 5** (Regret Bound). *Assume Assumption 2 to 5 holds. We have that*

$$\mathbb{E}_{\mathbb{P}(f)}[\text{Regret}(K)] \leq \tilde{O}\Big(\sqrt{H^2 T}$$

$$\cdot \sqrt{\log \mathcal{N}(\mathcal{F}, T^{-1/2}, \|\cdot\|_2)} \cdot \sqrt{\dim_E(\mathcal{F}, T^{-1/2})}\Big).$$

*where $\tilde{O}$ represents the order up to logarithm factors.*

For finite dimensional function class, $\log \mathcal{N}(\mathcal{F}, T^{-1/2}, \|\cdot\|_2)$ and $\dim_E(F, T^{-1/2}))$ should be scaled like $\mathrm{polylog}(T)$, hence our upper bound is sublinear in $T$. The proof is in Appendix C. Here we briefly sketch the proof idea.

*Proof Sketch.* We first construct an equivalent UCB algorithm (see Appendix B) and bound $\mathrm{Regret}(K)$ for it. Then by the conclusion from Russo and Van Roy [2013, 2014], Osband and Van Roy [2014], we can directly translate the upper bound on $\mathrm{Regret}(K)$ from UCB algorithm to an upper bound on $\mathbb{E}_{\mathbb{P}(f)}[\mathrm{Regret}(K)]$ of TS algorithm. We emphasize that the UCB algorithm is solely designed for analysis purpose.

With the optimism, we know for episode $k$, $V_0^*(s_0^k) \leq \tilde{V}_{0,k}^{\pi_k}(s_0^k)$, where $\tilde{V}_{h,k}^{\pi_k}$ is the value function of policy $\pi_k$ under the model $\tilde{f}_k$ introduced in the UCB algorithm. Hence, the regret at episode $k$ can be bounded by $\tilde{V}_{0,k}^{\pi_k}(s_0^k) - V_0^{\pi_k}(s_0^k)$, which is the value difference of the policy $\pi_k$ under the two models $\tilde{f}_k$ and $f^*$, that can be bounded by $\sqrt{\mathbb{E}\left[\sum_{h=0}^{H-1} \|f^*(s_h^k, a_h^k) - \tilde{f}_k(s_h^k, a_h^k)\|_2^2\right]}$ (see Lemma 9 for the details), which means when the estimated model $\hat{f}$ is close to the real model $f^*$, the policy obtained by planning on $\hat{f}$ will only suffer from a small regret. With Cauchy-Schwartz inequality, we only need to bound $\mathbb{E}\left[\sum_{k \in [K]} \sum_{h=0}^{H-1} \|f^*(s_h^k, a_h^k) - \tilde{f}_k(s_h^k, a_h^k)\|_2^2\right]$. This term can be handled via Lemma 13. With some additional technical steps, we can obtain the upper bound on $\mathrm{Regret}(K)$ for the UCB algorithm, and hence the upper bound on $\mathbb{E}_{\mathbb{P}(f)}[\mathrm{Regret}(K)]$ for the TS algorithm. $\square$

**Kernelized Non-linear Regulator** Notice that, for the linear function class $\mathcal{F} = \{\theta^\top \varphi(s, a) : \theta \in \mathbb{R}^{d_\varphi \times d}\}$ where $\varphi : \mathcal{S} \times \mathcal{A} \to \mathbb{R}^{d_\varphi}$ is a fixed and *known* feature map of certain RKHS, when the feature and the parameters are bounded, the logarithm covering number can be bounded by $\log \mathcal{N}(\mathcal{F}, \epsilon, \|\cdot\|_2) \lesssim d_\varphi \log(1/\epsilon)$, and the eluder dimension can be bounded by $\dim_E(\mathcal{F}, \epsilon) \lesssim d_\varphi \log(1/\epsilon)$ (see Appendix D for the detail, notice that we provide a tighter bound of the eluder dimension compared with the one derived in Osband and Van Roy [2014]). Hence, for linear function class, Theorem 5 can be translated into a regret upper bound of $\tilde{O}(H d_\varphi T^{1/2})$ for sufficiently large $T$, that matches the results of Kakade et al. [2020]. Moreover, for the case of linear bandits when $H = 1$, our bound can be

---

Note that, the RKHS here is the Hilbert space that contains $f(s, a)$ with the feature from some fixed and known kernel, It is different from the RKHS we introduced in Section 3, that contains $Q(s, a)$ with the feature $k(f(s, a), \cdot)$ where $k$ is the Gaussian kernel.

Note that $T$ in [Kakade et al., 2020] is the number of episodes, and $V_{\max}$ in [Kakade et al., 2020] can be viewed as $H^2$ when the per-step reward is bounded.

---

translated into a regret upper bound of $\tilde{O}(d_\varphi T^{1/2})$, that matches the lower bound [Dani et al., 2008] up to logarithmic terms.

**Compared with Kakade et al. [2020] and Osband and Van Roy [2014]** Our results have some connections with the results from Kakade et al. [2020] and Osband and Van Roy [2014]. However, in Kakade et al. [2020], the authors only considers the case when $\mathcal{F}$ only contains linear functions w.r.t some known feature map, which constrains its application in practice. We instead, consider the general function approximation, which makes our algorithm applicable for more complicated models like deep neural networks. Meanwhile, the regret bound from Osband and Van Roy [2014] depends on a global Lipschitz constant for the value function, which can be hard to quantify with either theoretical or empirical method. Instead, our regret bound gets rid of such dependency on the Lipschitz constant with the simulation lemma that carefully exploit the noise structure.

## 5 EXPERIMENTS

In this section, we study the empirical performance of SPEDE in the OpenAI MuJoCo control suite [Brockman et al., 2016]. We use the environments from MBBL [Wang et al., 2019], which varies slightly from the original environments in terms of modifying the reward function so its gradient w.r.t the states exists and introducing early termination (ET). Note that the set of environments contains various control and manipulation tasks, which are commonly used for benchmarking both model-free and model-based RL algorithms [*e.g.*, Kakade et al., 2020, Haarnoja et al., 2018]. As aforementioned, for practical implementation, our critic network consists of a representation network $\phi(\cdot)$ and a linear layer on the top. We follow the same procedure of Algorithm 1. Specifically, (1) for finding the optimal policy, we run an actor-critic algorithm (SAC); (2) we fix the representation network of the critic function $\phi(\cdot)$ and only update the linear layer on the top. We provide the full set of experiments in Appendix E.2 and the hyperparameter we use in Appendix E.6.

**Baselines** We compare our method with various model-based RL baselines: PETS [Chua et al., 2018] with random shooting (RS) optimizer, PETS with cross entropy method (CEM) optimizer and ME with TRPO policy optimizer [Kurutach et al., 2018]. Note that these are strong empirical baselines with many hand-tuned hyperparameters and engineering features (*e.g.*, ensemble of models). It is usually hard for any theoretically guaranteed model-based RL algorithm to match or surpass their performance [Kakade et al., 2020]. Another natural baseline is the successor feature [Dayan, 1993], which is one of the representative spectral features. We compare with the deep successor feature

---

Our code is available at https://sites.google.com/view/spede.

Table 1: Performance of SPEDE on various MuJoCo control tasks. All the results are averaged across 4 random seeds and a window size of 10K. Results marked with * is directly adopted from MBBL [Wang et al., 2019]. Our method achieves strong performance compared to pure empirical baselines (*e.g.*, PETS). We also compare SPEDE-REG which regularizes the critic using the model dynamics loss with several model-free RL method. SPEDE-REG significantly improves the performance of the SoTA method SAC.

| | Swimmer | Reacher | MountainCar | Pendulum | I-Pendulum |
|---|---|---|---|---|---|
| ME-TRPO* | 30.1±9.7 | -13.4±5.2 | -42.5±26.6 | **177.3±1.9** | -126.2±86.6 |
| PETS-RS* | 42.1±20.2 | -40.1±6.9 | -78.5±2.1 | 167.9±35.8 | -12.1±25.1 |
| PETS-CEM* | 22.1±25.2 | -12.3±5.2 | -57.9±3.6 | 167.4±53.0 | -20.5±28.9 |
| DeepSF | 25.5±13.5 | -16.8±3.6 | -17.0±23.4 | 168.6±5.1 | -0.2±0.3 |
| **SPEDE** | **42.6±4.2** | **-7.2±1.1** | **50.3±1.1** | 169.5±0.6 | **0.0±0.0** |
| PPO* | 38.0±1.5 | -17.2±0.9 | 27.1±13.1 | 163.4±8.0 | -40.8±21.0 |
| TRPO* | 37.9±2.0 | -10.1±0.6 | -37.2±16.4 | 166.7±7.3 | -27.6±15.8 |
| TD3* | 40.4±8.3 | -14.0±0.9 | -60.0±1.2 | 161.4±14.4 | -224.5±0.4 |
| SAC* | **41.2±4.6** | -6.4±0.5 | **52.6±0.6** | 168.2±9.5 | -0.2±0.1 |
| **SPEDE-REG** | 40.0±3.8 | **-5.8±0.6** | 40.0±3.8 | **168.5±4.3** | **0.0±0.1** |

| | Ant-ET | Hopper-ET | S-Humanoid-ET | Humanoid-ET | Walker-ET |
|---|---|---|---|---|---|
| ME-TRPO* | 42.6±21.1 | 4.9±4.0 | 76.1±8.8 | 72.9±8.9 | -9.5±4.6 |
| PETS-RS* | 130.0±148.1 | 205.8±36.5 | 320.9±182.2 | 106.9±106.9 | -0.8±3.2 |
| PETS-CEM* | 81.6±145.8 | 129.3±36.0 | 355.1±157.1 | 110.8±91.0 | -2.5±6.8 |
| DeepSF | 768.1±44.1 | 548.9±253.3 | 533.8±154.9 | 168.6±5.1 | 165.6±127.9 |
| **SPEDE** | **806.2±60.2** | **732.2±263.9** | **986.4±154.7** | **886.9±95.2** | **501.6±204.0** |
| PPO* | 80.1±17.3 | 758.0±62.0 | 454.3±36.7 | 451.4±39.1 | 306.1±17.2 |
| TRPO* | 116.8±47.3 | 237.4±33.5 | 281.3±10.9 | 289.8±5.2 | 229.5±27.1 |
| TD3* | 259.7±1.0 | 1057.1±29.5 | 1070.0±168.3 | 147.7±0.7 | **3299.7±1951.5** |
| SAC* | **2012.7±571.3** | 1815.5±655.1 | 834.6±313.1 | 1794.4±458.3 | 2216.4±678.7 |
| **SPEDE-REG** | **2073.1±119.7** | **2510.3±550.8** | **2710.3±277.5** | **3747.8±1078.1** | 2170.3±810.9 |

(DeepSF) [Kulkarni et al., 2016], and for a fair comparison, we only swap the representation objective of SPEDE with DeepSF and keep the other parts of the algorithm exactly the same.

**SPEDE: Performance with the Learned Representation** Following Algorithm 1, we are interested in how SPEDE performs when we conduct planning on top of the representation induced by the dynamics model in each episode. As most of the rigorously-justified representation learning algorithms are computationally intractable/inefficient, to demonstrate the effectiveness of representation used in SPEDE, we compare SPEDE with the deep successor features, which is one representative empirical representation learning algorithm. Moreover, as our method learning representation via fitting transition dynamics, to demonstrate the superiority of representation in planning, we compare our methods with the state-of-the-art model-based RL algorithms. We summarize the results of our method in Table 1. We see that our method achieves impressive performance comparing to model-based RL methods. Even in some hard environments that baselines fail to reach positive reward (*e.g.*, Mountain-Car, Walker-ET), SPEDE manage to achieve a reward of 52.6 and 501.6 respectively. We also evaluate our representation by comparing SPEDE to the usage of deep successor feature (DeepSF). Results show that on hard tasks like Humanoid and Walker, SPEDE manages to achieve 452.6 and 336.0 higher reward respectively.

**SPEDE-REG: Policy Optimization with SPEDE Representation Regularizer** In order to evaluate whether our assumption on linear MDP is valid in empirical settings and study whether such assumption can help improve the performance, we add our model dynamics representation objective as a regularizer in addition to the original SAC algorithm for learning the $Q$-function. Specifically, the algorithm SPEDE-REG consists of vanilla SAC objective with an additional loss putting constraints on the representation learned by the critic function, due to the intuition that the representation should satisfy the equivalent dynamics. We compare its performance with the vanilla SAC algorithm to show the benefits of dynamic representation. Results in Table 1 show that adding such constraint significantly improve the performance of SAC: on hard tasks like Hopper-ET, S-Humanoid-ET and Humanoid-ET, SPEDE-REG improves the performance of SAC by 694.8, 1875.7 and 2000.4.

**Ablations** We conduct ablations on: (1) What is the effect of the momentum parameter. (2) How does the number of random features affect the performance. Detailed results can be found at E.3.

**Performance Curves** To better understand how the sample complexity of our algorithm comparing to the prior model-based RL baselines, we plot the return versus environment steps in Figure 1. We see that comparing to prior model-based baselines, SPEDE enjoys great sample efficiency in these tasks. We want to emphasize that from MBBL [Wang et al., 2019], model-based methods already show significantly better sample efficiency compared to model-free methods (*e.g.*PPO/TRPO). We provide additional results in Appendix E.2.

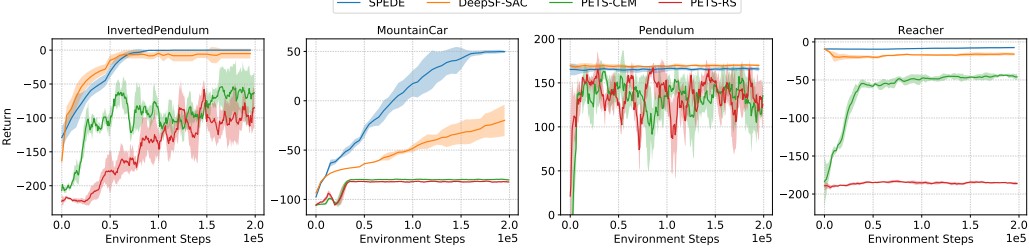

Figure 1: **Experiments on MuJoCo:** We show curves of the return versus the training steps for SPEDE and model-based RL baselines. Results show that in these tasks, our method enjoys better sample efficiency even compared to SoTA empirical model-based RL baselines.

**Discussion of the Results** We observe that in the environments with relatively simple dynamics (top row of Table 1), SPEDE achieves the SoTA among all the model-based and model-free RL algorithms. When the model dynamics of the environment become harder (bottom row of Table 1), the difference of the performance between the two approaches begin to enlarge. Interestingly, our SPEDE achieves strong results comparing to model-based approaches, while the joint learning SPEDE-REG outperforms model-free algorithm by a huge margin. The performance promotion of SPEDE indicates the importance on learning a good representation based on model dynamics and again shows the effectiveness of our approach in both settings. The performance gap might be caused by random feature approximation. To mitigate such approximation error, we also tried using MLP upon the learned representation, instead of linear form, which leads to better performances. Please refer to Appendix E.3 for details.

In fact, the differences in the SoTA usage of SPEDE in easy environments and difficult environments also reveals the important direction for our future work. The current rigorous representation learning methods, *e.g.*, Du et al. [2019], Misra et al. [2020], Agarwal et al. [2020a] and the proposed SPEDE, all rely on some model assumption. When the assumptions are satisfied, *e.g.*, Pendulum, Reacher, and others, our theoretically derived SPEDE variant works extremely well, even better than current SoTA. However, when the assumption is not fully satisfied, although the decoupled SPEDE achieves best performance among existing model-based RL and representation learning under fair comparison, the joint learned variant of SPEDE is more robust and promotes the current SoTA with significant margin. An interesting question is whether we can rigorously justify the regularized SPEDE, which we leave as our future work.

## 6 CONCLUSION

We introduce SPEDE, which, to the best of our knowledge, is the first provable and efficient representation learning algorithm for RL, by exploiting the benefits from noise. We provide thorough theoretical analysis and strong empirical

results, comparing to both model-free and model based RL, that demonstrates the effectiveness of our algorithm.

## ACKNOWLEDGEMENT

Cs. Sz. greatly acknowledges funding from NSERC, AMII and the Canada CIFAR AI Chair program. This project occurred under the Google-BAIR Commons at UC Berkeley.

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
