# OpenReview forum: "A Free Lunch from the Noise: Provable and Practical Exploration for Representation Learning"
_auai.org/UAI/2022/Conference — UAI 2022 Poster_

### Official Review · Reviewer_5ynz · 2022-04-08

**Q2(1) Originality/Novelty:** 3
**Q2(2) Significance/Impact:** 3
**Q2(3) Correctness/Technical Quality:** 3
**Q2(6) Clarity Of Writing:** 4
**Q6 Overall Score:** 7
**Q8 Confidence In Your Score:** 4

**Q1 Summary And Contributions:**

This paper focuses on representation learning in RL. It is motivated that the power of representation learning has not been fully exploited because of two unavoidable difficulties. This paper proposes an efficient algorithm SPEDE that can well handle the above difficulties by leveraging the noise structure. This paper also provides theoretical justifications in terms of regret and comprehensive evaluations to support their claims.

**Q2 Assessment Of The Paper:**

More detailed information regarding each of these aspects is given below:

**Q2(4) Quality Of Experiments (Optional):**

3: Good: The experimental evaluation is adequate, and the results convincingly support the main claims.

**Q2(5) Reproducibility:**

3: Good: Key resources (e.g., proofs, code, data) are available and key details (e.g., proofs, experimental setup) are sufficiently well-described for competent researchers to confidently reproduce the main results.

**Q3 Main Strengths:**

- This paper focuses on exploring efficient representation learning in RL whose motivation is clear and sufficient.
- This paper provides a clear description of the proposed method. The rationality of each part of the method is also clarified.
- This paper makes a comprehensive evaluation of different environments to verify the effectiveness of the proposed method.
- This paper is well organized and no obvious syntax errors are found.
- Detailed theoretical justifications are also provided.

**Q4 Main Weakness:**

- Some details are not given an explanation which makes readers somewhat confused.
- The theoretical justification relies on somewhat strong assumptions.

**Q5 Detailed Comments To The Authors:**

This paper is solid from the perspectives of theoretical results and experiments. Although I am not an expert on RL, I can understand the main ideas and work from this well-organized paper.

There are some suggestions:
- Some details should be illustrated to improve the readability. I am confused about how to compute the reward and why some results could be negative.
- There is a lot of blank space in Algorithm 2 and the formula is not aligned in line 4.


**Q7 Justification For Your Score:**

The motivation of this paper is clear and the logic is well-organized which is what I weigh most. The key observation transforms the problem into the estimation for f* which is reasonable and effective. The proposed method provides a provable efficient, and practical algorithm to achieve good representation learning which makes this paper contributes to this community. Moreover, extensive experiments and ablation studies suggest the applicability of the proposed methods in real-world scenarios.

**Q9 Complying With Reviewing Instructions:**

1: Yes.

---

### Official Review · Reviewer_NpVS · 2022-04-09

**Q2(1) Originality/Novelty:** 3
**Q2(2) Significance/Impact:** 3
**Q2(3) Correctness/Technical Quality:** 3
**Q2(6) Clarity Of Writing:** 3
**Q6 Overall Score:** 5
**Q8 Confidence In Your Score:** 3

**Q1 Summary And Contributions:**

This paper proposes Spectral Dynamics Embedding (SPEDE) algorithm for provable and efficient representation learning for RL, by exploiting the benefits from noise. Both theoretical analysis and empirical analysis are provided to show the effectiveness of SPEDE, comparing to both model-free and model based RL.

**Q2 Assessment Of The Paper:**

More detailed information regarding each of these aspects is given below:

**Q2(4) Quality Of Experiments (Optional):**

3: Good: The experimental evaluation is adequate, and the results convincingly support the main claims.

**Q2(5) Reproducibility:**

3: Good: Key resources (e.g., proofs, code, data) are available and key details (e.g., proofs, experimental setup) are sufficiently well-described for competent researchers to confidently reproduce the main results.

**Q3 Main Strengths:**

+ A SPEDE algorithm is proposed by exploiting the benefits from noise for provable and efficient representation learning for RL.
+ Both theoretical analysis and empirical analysis are provided to show the effectiveness of SPEDE.
+ The code for the proposed SPEDE is released.


**Q4 Main Weakness:**

- The reported results would be much stronger, if the latest methods published in 2021 are also included in Table 1.
- Different methods should be compared in terms of the time complexity, since ‘efficient’ is claimed in the Introduction.


**Q5 Detailed Comments To The Authors:**

See my comments in Q4.

**Q7 Justification For Your Score:**

A SPEDE algorithm is proposed for provable and efficient representation learning for RL. Both theoretical analysis and empirical analysis are provided. But the empirical analysis is insufficient to some extent.

**Q9 Complying With Reviewing Instructions:**

1: Yes.

---

### Official Review · Reviewer_891E · 2022-04-10

**Q2(1) Originality/Novelty:** 3
**Q2(2) Significance/Impact:** 3
**Q2(3) Correctness/Technical Quality:** 3
**Q2(6) Clarity Of Writing:** 1
**Q6 Overall Score:** 6
**Q8 Confidence In Your Score:** 1

**Q1 Summary And Contributions:**

The paper introduces an algorithm for efficiently performing representation learning for RL under some mild assumption about the noise distribution on the transition function. The paper appear to be theoretically founded and the experiments support the claims of the authors.

**Q2 Assessment Of The Paper:**

More detailed information regarding each of these aspects is given below:

**Q2(4) Quality Of Experiments (Optional):**

3: Good: The experimental evaluation is adequate, and the results convincingly support the main claims.

**Q2(5) Reproducibility:**

2: Fair: Key resources (e.g., proofs, code, data) are unavailable but key details (e.g., proof sketches, experimental setup) are sufficiently well-described for an expert to confidently reproduce the main results.

**Q3 Main Strengths:**

The paper presents a novel and important result.
Results have solid theoretical foundations.
Empirical evidence provides strong support for the effectiveness of the results.

**Q4 Main Weakness:**

The paper is very hard to read and understand. I am not well versed in the RL literature and I found most of the material too hard to comprehend. If on one side this can be attributed to my unfamiliarity with the subject, on the other side the authors do not make much effort to clarify the exposition.

The paper is poorly written. It is littered with grammatical errors and typos, which do not contribute to comprehensibility and compound with the clarity problem emphasized above.

**Q5 Detailed Comments To The Authors:**

As I admitted above, RL is not my main topic. Nevertheless, I think that the paper is too hard to read and I assume this is because it shows little effort to connect the technical results with the intuitions behind them. As a glaring example, nowhere in the paper, there is a description in lay terms of the connection between the theoretical results and representation learning. I put some effort into trying to make that connection, but I failed. This should be something that anyone, regardless of how little background he/she has, should be able to understand.

As I mentioned, in addition to the main narration problems, the paper is also littered with grammatical errors and typos. I suggest thoroughly reviewing the paper (ask the help of a native speaker if necessary) to correct these problems. It is useful to keep in mind that these communication problems are present also in the more technical parts. For instance, you should be careful to introduce each and every piece of notation you use, especially if it is not common or have a different meaning in nearby research fields (e.g., you never introduce the [n] notation). Also, try to give the intuition behind each symbol and each formula, so that formulas clarify and make precise the reasoning, but the main argument can be also understood at a high abstraction level.

**Q7 Justification For Your Score:**

As I mentioned the paper appears to be technically sound and the experiments provide solid support for the approach, but the paper is not very well written. If this was a journal submission I would probably ask for a revision that fixes these communication problems. Since this cannot be done and I would find it unfair to reject the paper for this reason only, I rated the paper as a weak accept, but I hope the authors take the chance to fix these problems before publication.

**Q9 Complying With Reviewing Instructions:**

1: Yes.

---

### Official Review · Reviewer_Hy5E · 2022-04-11

**Q2(1) Originality/Novelty:** 2
**Q2(2) Significance/Impact:** 2
**Q2(3) Correctness/Technical Quality:** 2
**Q2(6) Clarity Of Writing:** 2
**Q6 Overall Score:** 6
**Q8 Confidence In Your Score:** 2

**Q1 Summary And Contributions:**

This paper introduces the so-called Spectral Dynamics Embedding (SPEDE), which leverages the structure of noise thanks to the MDP formulation in order to provide an efficient and simple algorithm for representation learning in reinforcement learning


**Q2 Assessment Of The Paper:**

More detailed information regarding each of these aspects is given below:

**Q2(4) Quality Of Experiments (Optional):**

2: Fair: The experimental evaluation is weak: important baselines are missing, or the results do not adequately support the main claims.

**Q2(5) Reproducibility:**

2: Fair: Key resources (e.g., proofs, code, data) are unavailable but key details (e.g., proof sketches, experimental setup) are sufficiently well-described for an expert to confidently reproduce the main results.

**Q3 Main Strengths:**

The proposed method seems interesting. The algorithm is relatively simple, and theoretical results allow to provide some guarantees on the regret bound.


**Q4 Main Weakness:**

We didn’t find major weaknesses. See detailed comments for some raised issues.


**Q5 Detailed Comments To The Authors:**

The authors provide some sentences that are not clear in presenting the contributions of this paper. For example, it is said that“Our most important observation is that, the density of isotropic Gaussian distribution can be expressed as the inner product of two feature maps, thanks to the reproducing property and the random Fourier transform of the Gaussian kernel”. However, these properties have been been known from a while by researchers in machine learning, such as kernel-based methods and Gaussian processes.

It turns out that several sentences are not readable, such as “We found this prevent updating the representation of too fast”

There are many spelling and grammatical errors, such as “on an reliable estimation”, “This equivalency immediately overcome”, “demonstrate the superior of”, “rely on explicitly decomposion of”, “exploratory polices”, “establishes the equivalency among”, “The regret of the optimstic”, “require an planning”, “might be computational intractable”, “Notce that”, “perform a dynamic programming style algorithm that calculate”, “for sufficient large”, “even we have large”, “demonstrate the superior of representation”, “The performance gap might caused by random”, “is more robust and promote the”, … and many missing determiner “a/the”

The paper should be proofread, including the appendices that also suffer from spelling and grammatical issues.


**Q7 Justification For Your Score:**

Our overall assessment is based on all the aforementioned comments.


**Q9 Complying With Reviewing Instructions:**

1: Yes.

---

### Decision · Program_Chairs · 2022-05-15

**Decision:**

Accept (Poster)

**Comment:**

Meta Review: The paper considers an algorithm with provable regret bounds for an MDP where the transition function has Gaussian noise. The main theoretical advance is the choice of an appropriate "basis" (or "representation" as the authors call it), in which to run the planning algorithm, based on the Gaussanity of the noise. Precisely, one uses the eigenfunctions of an appropriate Gaussian kernel as a basis.

The paper has a nice idea, though as the reviewers pointed out, it has issues with clarity of the writing (which should be addressed, should the paper be accepted).  Assumptions/setup is kind of sprinkled throughout (some in preliminaries in section 2, some in section 4); the comparison to prior work is a bit hard to keep track of (sprinkled throughout pages 4, 5, 7, and it's not so clear what comparisons are essential). The stated "high level question" is a bit too ambitious ("How to design provably efficient and practical algorithm for representation learning in RL?") for how specific the result is --- it's a bit hard to argue this paper says something very general about representation learning, as the results are very specific to the Gaussian setting considered.

Some reviewers also raised issues about the experimental evaluation, though the paper is predominantly making a theoretical contribution, so this is less of an issue in the meta-reviewers opinion.